# Identification and Characterization of *Colletotrichum* Species Causing Tea-Oil Camellia (*Camellia oleifera* C.Abel) Anthracnose in Hainan, China

**Hui Zhu** [1,2,*] **and Chaozu He** [1]

[1] Hainan Key Laboratory for Sustainable Utilization of Tropical Bioresources, College of Tropical Crops, Hainan University, Haikou 570228, China; czhe@hainanu.edu.cn

[2] Coconut Research Institute, Chinese Academy of Tropical Agricultural Sciences, Wenchang 571339, China

\* Correspondence: zhuhui@catas.cn

**Abstract:** *Camellia oleifera* C.Abel, commonly known as tea-oil camellia, is a type of significant woody oil crop that is widely cultivated in southern China. During 2017–2021, a fungal foliar disease was detected in routine surveys in Hainan. However, diseases of tea-oil camellia are seldom reported in Hainan. In this study, 51 *Colletotrichum* spp. isolates were obtained from the symptomatic samples of tea-oil camellia, collected from three production sites located in Hainan. A polyphasic approach was applied to distinguish *Colletotrichum* species. All 51 isolates were primarily characterized morphologically, and six gene regions, including an internally transcribed space of ribosomal DNA (ITS), chitinsynthase (CHS-1), β-tubulin (TUB), actin (ACT), glyceraldehyde-3-phosphate dehydrogenase (GAPDH), and manganese-superoxide (SOD2), were sequenced for each isolate. By combining morphological characterization with multilocus sequence analysis (MLSA) based on the six genes, the fungal isolates were identified, representing three *Colletrotrichum* species: *C. fructicola*, *C. siamense*, and *C. cordylinicola*. The most predominant species was *C. fructicola*. In pathogenicity tests on the tea-oil camellia cultivar (Reyan1), all collected isolates were pathogenic on tea-oil camellia leaves and were reisolated from symptomatic leaves. *Colletotrichum fructicola* was the most aggressive species on the attached leaves. This is the first report of *C. cordylinicola* affecting tea-oil camellia anthracnose worldwide. These results will improve our understanding of the pathogens and provide important insights on the diagnosis and efficient disease management of tea-oil camellia anthracnose.

**Keywords:** tea-oil camellia; *Camellia oleifera*; anthracnose; *Colletotrichum*; multilocus sequence analysis; etiology

## 1. Introduction

Tea-oil camellia, *Camellia oleifera* C.Abel, is a member of the Theaceae family. It is originated in China and is the main crop used for the production of edible oil in this country [1,2]. It is primarily grown in China's central and southern regions, and a small amount is distributed in Southeast Asia and Japan due to its strong environmental adaptability. Camellia oil, derived from camellia fruits, has a high fatty acid content and profitable microelements and is considered one of the healthiest vegetable oils [3]. Camellia oil also has unique medicinal functions, such as enhancing human immunity and preventing cardiovascular diseases [4]. In 2020, the planting area of tea-oil camellia reached almost 4,533,333 hectares, and the total output of camellia oil reached 627,000 tons in China [5]. Therefore, tea-oil camellia is not only capable of improving human health but also is an economically important crop providing a main source of livelihood for many farmers in China [6].

However, problems caused by diseases are a key factor limiting tea-oil camellia yield. In China, tea-oil camellia can be infected by over 30 species of fungi [7], among which *Colletotrichum* species causing anthracnose are the most important pathogens in most tea-oil

camellia production areas. *Colletotrichum* spp. usually exhibit strong virulence to hosts, rapid rate of spread, and in general are considered to be destructive plant pathogens [8]. To date, many plant taxa have been reported as hosts to *Colletotrichum* spp. Li et al. [9] reported a fruit anthracnose caused by *C. fructicola* on *Passiflora edulis* and Li et al. [10] reported that both *C. siamense* and *C. fructicola* were pathogenic to luffa sponge gourd (*Luffa cylindrica*). Qiao et al. [11] reported that *C. menglaense* may be a potential pathogen of strawberry fruit.

The symptoms of tea-oil camellia anthracnose include abscission of fruits, buds, and leaves; death of branches; and sometimes death of an entire plant. The disease can advance rapidly and result in severe economic losses. An almost 40% yield loss and a 20%–40% fruit drop could be caused by the disease, and sometimes, the whole plant can be killed [12,13], leading to significant yield and economic losses, badly affecting the supply of edible oil in China.

The identification of pathogen species is the basis of plant disease monitoring and control. The species identification of the genus *Colletotrichum* has for the long time been conducted on the basis of morphological and biological characteristics [14] such as conidial, conidiophore, acervuli morphology [15], and benomyl sensitivity [16]. However, with the development of *Colletotrichum* classification research, the traditional techniques were insufficient to differentiate *Colletotrichum* species due to the phenotypic variations in different geographical distributions and environmental situations [17]. Therefore, molecular approaches are being increasingly applied for their identification [14,18]. The phylogenetic analysis of internally transcribed space of ribosomal DNA (ITS) has been previously utilized for this purpose [15], while many closely related *Colletotrichum* species still could not be distinguished by using only ITS [19–21]. Presently, phylogenetic analysis based on multiple genes, including ITS, glyceraldehyde-3-phosphate dehydrogenase (GAPDH), actin (ACT), chitinsynthase (CHS-1), β-tubulin (TUB), and histone H3 (HIS), has become widely applied as an efficient tool for fungal species identification and delimitation [22–25]. The *C. gloeosporioides* species complex (CGSC) has been classified into 22 species using multiloci (eight genes) phylogeny [21]. A few years later, 37 species had already been distinguished in the CGSC by multilocus phylogenetic analyses [26].

A study by Li et al. [27] showed that the causal pathogens of tea-oil camellia anthracnose belong to the CGSC. Recently, outbreaks of tea-oil camellia anthracnose have been reported in southern China. By using morphological characteristics and multilocus phylogenetic analyses, several *Colletotrichum* species infecting tea-oil camellia were identified as *C. boninense* in Jiangxi Province [28], *C. fructicola* and *C. siamense* in Guangdong and Hunan Provinces [29,30], *C. gloeosporioides* in Guangxi and Hunan Provinces [30,31], *C. horii* and *C. camelliae* in Hunan Province [30], and *C. fructicola* and *C. camelliae* in Fujian Province [32]. Furthermore, *C. karstii*, *C. aenigma*, and *C. nymphaeae* were also identified as the causal agents of tea-oil camellia anthracnose [33–35]. Hainan is the southernmost province in China with a typical tropical climate. However, the pathogen diversity of tea-oil camellia has never previously been investigated systematically in Hainan.

In this study, during field surveys between 2017 and 2021, symptoms of anthracnose on tea-oil camelia leaves were consistently observed in plantations in Wenchang, Qiongzhong, and Wuzhishan of Hainan Province, China. As the disease progressed, lesions became irregular or circular dark-brown spots that expanded and coalesced into larger patches under favorable environmental conditions. The anthracnose and subsequent leaf mortality have significantly affected the ornamental and economic values of tea-oil camellia. The main aim of the present study was to identify the causal agents of tea-oil camellia anthracnose in Hainan Province, China, by utilizing a polyphasic approach, including morphological characterization, pathogenicity assays and multilocus phylogenetic analyses of the collected *Colletotrichum* isolates. Correct identification of the disease-causing agents will provide a basis for creating more targeted prevention and control techniques against tea-oil camellia anthracnose.

## 2. Materials and Methods

### 2.1. Field Survey, Sample Collection and Fungal Isolation

Field surveys and sample collection were conducted during 2017–2021 in three major tea-oil camellia plantations in Hainan Province, China. The three plantations are located in Wenchang (19.54° N, 110.77° E), Qiongzhong (18.97° N, 109.86° E), and Wuzhishan (18.82° N, 109.54° E). All three plantations have a tropical climate with intermittent rainfall. During field surveys, premature leaf fall and leaf spot symptoms were observed. The disease symptoms were distributed widely and could be spotted easily across all plantations during the field surveys. Thirty symptomatic leaves from each plantation were collected from randomly selected trees. One to three samples per tree were taken and a total of 185 leaf samples with symptoms of anthracnose were collected. The spots on the collected leaves were sheared into 3–5 mm pieces by a surgical scissor. The leaf pieces were first disinfected with 75% ethanol for 30 s, then disinfected with 1.5% sodium hypochlorite for 60 s; after being washed four times with sterilized ultrapure water, the pieces were placed onto potato dextrose agar (PDA) plates amended with streptomycin sulfate (0.5 mg/L). All the PDA plates were incubated for 5 days at 25 °C. Agar plugs (5 mm diameter) cut out at the margin of the emerging mycelium were transferred to new PDA plates and purified further by single-spore cultures after sporulation [36]. Purified isolates were preserved on PDA slants at 5 °C for further use. A total of 51 *Colletotrichum* isolates were obtained, and detailed information about the isolates is shown in Table 1.

**Table 1.** List of all *Colletotrichum* isolates obtained from tea-oil camellia (*Camellia oleifera* C.Abel) anthracnose.

| Isolate Code | Location | Year | Isolate Code | Location | Year |
|---|---|---|---|---|---|
| yc01 | Wenchang | 2018 | yc27 | Qiongzhong | 2021 |
| yc02 | Qiongzhong | 2019 | yc28 | Qiongzhong | 2020 |
| yc03 | Wenchang | 2017 | yc29 | Qiongzhong | 2020 |
| yc04 | Wenchang | 2020 | yc30 | Wenchang | 2017 |
| yc05 | Qiongzhong | 2018 | yc31 | Wenchang | 2020 |
| yc06 | Qiongzhong | 2021 | yc32 | Wenchang | 2021 |
| yc07 | Wuzhishan | 2021 | yc33 | Wenchang | 2020 |
| yc08 | Wuzhishan | 2020 | yc34 | Wenchang | 2018 |
| yc09 | Wuzhishan | 2019 | yc35 | Wenchang | 2021 |
| yc10 | Wenchang | 2021 | yc36 | Wenchang | 2021 |
| yc11 | Wenchang | 2021 | yc37 | Wenchang | 2019 |
| yc12 | Wuzhishan | 2018 | yc38 | Wenchang | 2021 |
| yc13 | Wuzhishan | 2021 | yc39 | Wenchang | 2020 |
| yc14 | Wuzhishan | 2021 | yc40 | Wenchang | 2018 |
| yc15 | Wuzhishan | 2021 | yc41 | Wenchang | 2021 |
| yc16 | Qiongzhong | 2019 | yc42 | Wenchang | 2020 |
| yc17 | Qiongzhong | 2019 | yc43 | Wenchang | 2021 |
| yc18 | Qiongzhong | 2021 | yc44 | Wenchang | 2019 |
| yc19 | Qiongzhong | 2017 | yc45 | Wenchang | 2021 |
| yc20 | Qiongzhong | 2020 | yc46 | Wuzhishan | 2021 |
| yc21 | Qiongzhong | 2020 | yc47 | Wuzhishan | 2021 |
| yc22 | Qiongzhong | 2018 | yc48 | Wuzhishan | 2020 |
| yc23 | Qiongzhong | 2018 | yc49 | Wuzhishan | 2019 |
| yc24 | Qiongzhong | 2021 | yc50 | Wuzhishan | 2021 |
| yc25 | Qiongzhong | 2019 | yc51 | Wuzhishan | 2017 |
| yc26 | Qiongzhong | 2021 | — | — | — |

### 2.2. Cultural and Morphological Characterization

All the 51 *Colletotrichum* isolates were incubated on PDA plates at 25 °C for 5 days. A 5 mm diameter mycelial plug cut out from the boundary of developing colonies of each isolate was transferred to a fresh PDA plate and incubated at 25 °C for 7 days. All the isolates were inspected daily and the experiment was replicated four times. Morphological and cultural characteristics, such as the shape, color and pigmentation of each isolate and

the color of conidial masses, were recorded on the fifth day post-inoculation. The colony diameter of each isolate has been recorded daily for 7 days and used to determine the hyphal growth rate. For each isolate, conidia ($n = 50$) were chosen at random to assess their length, width, and shape. Appressoria induction was conducted using a slide culture technique [18]. The shape, length, and width of the appressoria ($n = 50$) of each isolate were recorded. The morphological characteristics of conidia and appressoria were determined and photographed using a Leica DMIL inverted microscope (Leica, Wetzlar, Germany).

### 2.3. Pathogenicity Assay

The pathogenicity assay for each of the 51 isolates was conducted using a drop-inoculation method [37] with slight modifications. All the *Colletotrichum* isolates were inoculated on PDA plates at 25 °C for 6 days. The spore suspension of each isolate has been prepared using sterilized ultrapure water. Spore concentration was measured with a hemocytometer, and adjusted to $2 \times 10^6$ conidia/mL. Healthy tea-oil camellia leaves and fruits (cv. Reyan1) were selected, washed with flowing tap water, dried at room temperature, and disinfected with 70% ethanol before the pathogenicity assay. All the collected leaves and fruits were wounded with three piercing wounds in the mid-region using a disinfectant insect needle. The conidia suspension of each isolate was dropped onto the leaves and fruits. The same approach was used to inoculate the controls with sterile water. After inoculation, all the leaves and fruits were placed in sealed plastic chambers at $26 \pm 1$ °C for 3 days. The humidity was maintained above 90%. The pathogenicity assay was performed with ten leaves and fruits per isolate and replicated twice. Symptomatic leaves and fruits were collected for the re-isolation of the *Colletotrichum* isolates. Pathogenicity was certified by checking the morphological characteristics and ITS sequence of each reisolated fungus, completing Koch's postulates confirmation. Infection incidence was calculated to determine the pathogenicity of each isolate.

### 2.4. Fungal DNA Extraction, PCR Amplification, and DNA Sequencing

After 6 days of incubation at 25 °C, the total genomic DNA of each *Colletotrichum* isolates was extracted for PCR amplification using an SDS extraction method described by Hossain [38]. The concentration of the extracted DNA was determined by using a spectrophotometer and adjusted to 15 ng/L with ultrapure water. The DNA of each isolate was stored at $-20$ °C and used as templates for PCR amplification.

For PCR amplification, six loci genes, including ITS, TUB, ACT, GAPDH, CHS-1, and superoxide dismutase gene (SOD2), were amplified by applying the ITS1-1F/ITS4, T1/T2, ACT512F/ACT783R, GDF/GDR, CHS-79F/CHS-345R, and SODglo2-F/SODglo2-R primers, respectively (Table 2). A total 25 μL reaction volume, including template DNA (2 μL), dNTPs (2 μL), reaction buffer (2.5 μL), each primer (1.5 μL), Taq polymerase (0.25 μL), and ultrapure water (15.25 μL), was used for PCR amplification. PCR amplification was conducted by using a TProfessional Thermocycler (Biometra, Göttingen, Germany). Thermo-cycling parameters used for the ITS region included the initial denaturation step at 95 °C for 4 min, followed by 95 °C for 40 s, 52 °C for 40 s, and 72 °C for 60 s, reduplicating for 30 cycles, and a last extension step at 72 °C for 8 min. The thermo-cycling parameters used for other genes were consistent with those of ITS but were adjusted by applying an annealing temperature of 55 °C for TUB, 58 °C for ACT and CHS-1, 60 °C for GAPDH, and 54 °C for SOD2.

**Table 2.** Primer sequences used for the molecular identification of the *Colletotrichum* isolates in this study.

| Number | Target Fragment | Primer Name | Primer Sequence (5′-3′) | Source |
|---|---|---|---|---|
| 1 | Internally transcribed space of ribosomal DNA (ITS) | ITS1-1F<br>ITS4 | CTTGGTCATTTAGAGGAAGTAA<br>TCCTCCGCTTATTGATATGC | Gardes et al. [39] |
| 2 | β-tubulin (TUB) | T1<br>T2 | AACATGCGTGAGATTGTAAGT<br>TAGTGACCCTTGGCCCAGTTG | O'Donnell et al. [40] |
| 3 | Chitinsynthase (CHS-1) | CHS-79F<br>CHS-345R | TGGGGCAAGGATGCTTGGAAGAAG<br>TGGAAGAACCATCTGTGAGAGTTG | Carbone et al. [41] |
| 4 | Actin (ACT) | ACT512F<br>ACT783R | ATGTGCAAGGCCGGTTTCGC<br>TACGAGTCCTTCTGGCCCAT | Carbone et al. [41] |
| 5 | Glyceraldehyde-3-phosphate dehydrogenase (GAPDH) | GDF<br>GDR | GCCGTCAACGACCCCTTCATTGA<br>GGGTGGAGTCGTACTTGAGCATGT | Peres et al. [42] |
| 6 | Manganese-superoxide (SOD2) | SODglo2-F<br>SODglo2-R | CAGATCATGGAGCTGCACCA<br>TAGTACGCGTGCTCGGACAT | Moriwaki et al. [43] |

The results of each PCR amplification were examined through 1% (*w/v*) agarose electrophoresis. PCR products were extracted and purified using a spin column DNA extraction kit (Sangon Biotech, Shanghai, China), in accordance with the producer's guidance. Then, the purified PCR products were sent to Sangon Biotech, Shanghai, China for DNA sequencing in both the forward and reverse directions. The sequences of the six regions were submitted to GenBank database and the associated accession numbers are as shown in Table S1.

*2.5. Phylogenetic Analysis*

The sequences of 43 reference species of the CGSC and two outgroups were selected from the GenBank database for conducting sequence alignment and phylogenetic analyses (Table 3). Six gene sequences, including ITS, TUB, CHS-1, GAPDH, ACT, and SOD2, were used to perform a phylogenetic analysis of *Colletotrichum* isolates. Each gene sequence of all isolates was first aligned by applying ClustalX [44] and manually edited and combined by using MEGA 7 [45] when necessary. To ensure that all sequences were of the same length, gaps in the alignment were considered as gap data.

All the six gene sequences were concatenated by an order of ITS–TUB–ACT–GAPDH–CHS-1–SOD2 to a length of 2524 bp. The boundary of each gene in the matrix was as follows: ITS: 1–593, TUB: 594–1292, GAPDH: 1293–1569, ACT: 1570–1851, CHS-1: 1852–2150, and SOD2: 2151–2524. Bayesian inference (BI) and maximum likelihood (ML) were used to conduct phylogenetic analyses based on the concatenated sequences. For BI analyses, Mrmodeltest 2.2 [46] was applied to determine the best-fit nucleotide substitution model of each gene. The substitution models for ITS, TUB, GAPDH, ACT, CHS-1, and SOD2 genes used for the Bayesian study were SYM + I + G, HKY + I, HKY + I, HKY + I, SYM + G, and GTR + I + G, respectively (Table 4). MrBayes 3.1.2 [47] was launched to construct phylogenetic trees. In the Markov Chain Monte Carlo (MCMC) analysis, four chains were run for 10,000,000 generations. Trees were sampled every 1000 generations. When the average standard deviation of split frequencies fell below 0.01, the BI analyses stopped. In each analysis, the first 25% trees were abandoned as the burn-in phase and the posterior probabilities were calculated based on the last 75%. MEGA 7 was employed to conduct the ML analysis and the distance matrix was based on the Kimura two-parameter distance method. A bootstrap analysis with 1000 replicates was used to calculate the confidence value in the tree. The phylogenetic tree was viewed in Treeview [48]. The outgroup species used in this study were *C. hippeastri* CBS 241.78 and *C. boninense* ICMP17904.

**Table 3.** Reference isolates of *Colletotrichum* species used for multilocus phylogenetic analyses from the GenBank database.

| Species Name | Culture | Country | Host | Accession Number | | | | | |
|---|---|---|---|---|---|---|---|---|---|
| | | | | ITS | TUB | CHS-1 | ACT | GAPDH | SOD2 |
| *C.aenigma* | ICMP 18608 * | Israel | *Persea americana* | JX010244 | JX010389 | JX009774 | JX009443 | JX010044 | JX010311 |
| | ICMP 18686 | Japan | *Pyrus pyrifolia* | JX010243 | JX010390 | JX009789 | JX009519 | JX009913 | JX010312 |
| *C. boninense* | ICMP17904 *, CBS 123755 | Japan | *Crinum asiaticum* | JX010292 | JQ005588 | JX009827 | JX009583 | JX009905 | — |
| *C. alatae* | CBS 304.67 *, ICMP 17919 | India | *Dioscorea alata* | JX010190 | JX010383 | JX009837 | JX009471 | JX009990 | JX010305 |
| *C. alienum* | IMI 313842, ICMP 18691 | Australia | *Persea americana* | JX010217 | JX010385 | JX009754 | JX009580 | JX010018 | JX010307 |
| | ICMP 12071 * | New Zealand | *Malus domestica* | JX010251 | JX010411 | JX009882 | JX009572 | JX010028 | JX010333 |
| *C. aotearoa* | ICMP 18532 | New Zealand | *Vitex lucens* | JX010220 | JX010421 | JX009764 | JX009544 | JX009906 | JX010338 |
| | ICMP 17324 | New Zealand | *Kunzea ericoides* | JX010198 | JX010418 | JX009770 | JX009538 | JX009991 | JX010344 |
| *C. asianum* | IMI 313839, ICMP18696 | Australia | *Mangifera indica* | JX010192 | JX010384 | JX009753 | JX009576 | JX009915 | JX010306 |
| | ICMP18580 *, CBS 130418 | Thailand | *Coffea arabica* | JX010196 | JX010406 | JX009867 | JX009584 | JX010053 | JX010328 |
| *C. cordylinicola* | MFLUCC090551 *, ICMP18579 | Thailand | *Cordyline fruticosa* | JX010226 | JX010440 | JX009864 | JX009586 | JX009975 | JX010361 |
| *C. clidemiae* | ICMP 18706 | USA | *Vitis* sp. | JX010274 | JX010439 | JX009777 | JX009476 | JX009909 | JX010353 |
| | ICMP18658 * | USA, Hawaii | *Clidemia hirta* | JX010265 | JX010438 | JX009877 | JX009537 | JX009989 | JX010356 |
| *C. fructicola* | ICMP18613 | Israel | *Limonium sinuatum* | JX010167 | JX010388 | JX009772 | JX009491 | JX009998 | JX010310 |
| | CBS 125395, ICMP18645 | Panama | *Theobroma cacao* | JX010172 | JX010408 | JX009873 | JX009543 | JX009992 | JX010330 |
| *C. gloeosporioides* | IMI356878 *, ICMP17821, CBS112999 | Italy | *Citrus sinensis* | JX010152 | JX010445 | JX009818 | JX009531 | JX010056 | JX010365 |
| | ICMP12938 | New Zealand | *Citrus sinensis* | JX010147 | — | JX009746 | JX009560 | JX009935 | — |
| *C. hippeastri* | CBS 241.78, ICMP17920 | Netherlands | *Hippeastrum* sp. | JX010293 | — | JX009838 | JX009485 | JX009932 | — |
| *C. horii* | ICMP12942 | New Zealand | *Diospyros kaki* | GQ329687 | JX010375 | JX009748 | JX009533 | GQ329685 | JX010296 |
| | ICMP17968 | China | *Diospyros kaki* | JX010212 | JX010378 | JX009811 | JX009547 | GQ329682 | JX010300 |
| *C. kahawae* subsp. *ciggaro* | ICMP18539 * | Australia | *Olea europaea* | JX010230 | JX010434 | JX009800 | JX009523 | JX009966 | JX010346 |
| | IMI 359911, ICMP17931,CBS12988 | Switzerland | *Dryas octopetala* | JX010236 | JX010428 | JX009832 | JX009475 | JX009965 | JX010354 |
| *C. kahawae* subsp. *ciggaro* | CBS 237.49 *, ICMP17922 | Germany | *Hypericum perforatum* | JX010238 | JX010432 | JX009840 | JX009450 | JX010042 | JX010355 |
| | CBS 124.22 *, ICMP19122 | USA | *Vaccinium* sp. | JX010228 | JX010433 | JX009902 | JX009536 | JX009950 | JX010367 |
| *C. kahawae* subsp. *kahawae* | CBS982.69, ICMP17915 | Angola | *Coffea arabica* | JX010234 | JX010435 | JX009829 | JX009474 | JX010040 | JX010352 |
| | IMI 361501, ICMP17905 | Cameroon | *Coffea arabica* | JX010232 | JX010431 | JX009816 | JX009561 | JX010046 | JX010349 |
| *C. musae* | CBS116870 *, ICMP19119 | USA | *Musa* sp. | JX010146 | HQ596280 | JX009896 | JX009433 | JX010050 | JX010335 |
| | IMI 52264, ICMP17817 | Kenya | *Musa sapientum* | JX010142 | JX010395 | JX009815 | JX009432 | JX010015 | JX010317 |
| *C. nupharicola* | CBS 469.96, ICMP17938 | USA | *Nupharlutea* subsp. *polysepala* | JX010189 | JX010397 | JX009834 | JX009486 | JX009936 | JX010319 |
| | CBS 470.96 *, ICMP18187 | USA | *Nupharlutea* subsp. *polysepala* | JX010187 | JX010398 | JX009835 | JX009437 | JX009972 | JX010320 |
| *C. psidii* | CBS 145.29 *, ICMP19120 | Italy | *Psidium* sp. | JX010219 | JX010443 | JX009901 | JX009515 | JX009967 | JX010366 |
| *C. queenslandicum* | ICMP1778 * | Australia | *Carica papaya* | JX010276 | JX010414 | JX009899 | JX009447 | JX009934 | JX010336 |
| | ICMP18705 | Fiji | *Coffea* sp. | JX010185 | JX010412 | JX009890 | JX009490 | JX010036 | JX010334 |
| *C. salsolae* | ICMP19051 * | Hungary | *Salsola tragus* | JX010242 | JX010403 | JX009863 | JX009562 | JX009916 | JX010325 |
| | CBS 119296, ICMP18693 | Hungary | *Glycine max* | JX010241 | — | JX009791 | JX009559 | JX009917 | — |
| *C. siamense* | ICMP12567 | Australia | *Perseaa mericana* | JX010250 | JX010387 | JX009761 | JX009541 | JX009940 | JX010309 |
| | ICMP18121 | Nigeria | *Dioscor earotundata* | JX010245 | JX010402 | JX009845 | JX009460 | JX009942 | JX010324 |
| *C. theobromicola* | MUCL42295, ICMP17958, CBS 124250 | Australia | *Stylosanthes guianensis* | JX010291 | JX010381 | JX009822 | JX009498 | JX009948 | JX010303 |
| | ICMP17895 | Mexico | *Annona diversifolia* | JX010284 | JX010382 | JX009828 | JX009568 | JX010057 | JX010304 |

**Table 3.** *Cont.*

| Species Name | Culture | Country | Host | Accession Number | | | | | |
|---|---|---|---|---|---|---|---|---|---|
| | | | | ITS | TUB | CHS-1 | ACT | GAPDH | SOD2 |
| *C. ti* | ICMP 5285 | New Zealand | *Cordyline australis* | JX010267 | JX010441 | JX009897 | JX009553 | JX009910 | JX010363 |
| | ICMP 4832 * | New Zealand | *Cordyline* sp. | JX010269 | JX010442 | JX009898 | JX009520 | JX009952 | JX010362 |
| *C. tropicale* | MAFF 239933, ICMP 18672 | Japan | *Litchi chinensis* | JX010275 | JX010396 | JX009826 | JX009480 | JX010020 | JX010318 |
| | CBS 124949 *, ICMP 18653 | Panama | *Theobroma cacao* | JX010264 | JX010407 | JX009870 | JX009489 | JX010007 | JX010329 |
| *C. xanthorrhoeae* | BRIP 45094 *, ICMP 17903 | Australia | *Xanthorrhoea preissii* | JX010261 | JX010448 | JX009823 | JX009478 | JX009927 | JX010369 |
| | IMI 350817a, ICMP 17820 | Australia | *Xanthorrhoea* sp. | JX010260 | — | JX009814 | JX009479 | JX010008 | — |

* Represent ex-holotype or ex-epitype cultures.

**Table 4.** Best-fit evolutionary model selection for each gene.

| Gene Datasets | ITS | TUB | GAPDH | ACT | CHS-1 | SOD2 |
|---|---|---|---|---|---|---|
| Best-fit evolutionary model | SYM + I + G | HKY + I | HKY + I | HKY + I | SYM + G | GTR + I + G |

## 3. Results

### 3.1. Disease Symptom Characteristics

Based on field observations, disease symptoms were recorded. Typical anthracnose symptoms usually appeared in July and August, when the hot and humid weather is suitable for disease progression. Initially, water-soaked, round or ovoid, light-yellow spots 1–3 mm in diameter were observed on the diseased leaves' surface. Seriously infected leaves often had dense spots all over the entire leaves (Figure 1A) with round or irregular yellow-brown lesions on the reverse side (Figure 1B). With disease progression, small spots gradually expanded or consolidated to ovoid or irregular bigger spots. The margin of the lesions changed to brown and the central part subsequently changed to dark brown, with black particles (acervuli) appearing on the surface in the humid climate (Figure 1C). At an advanced stage of pathogen infection, the diseased leaves always shrunk and shed prematurely (Figure 1D), resulting in serious economic losses to local farmers.

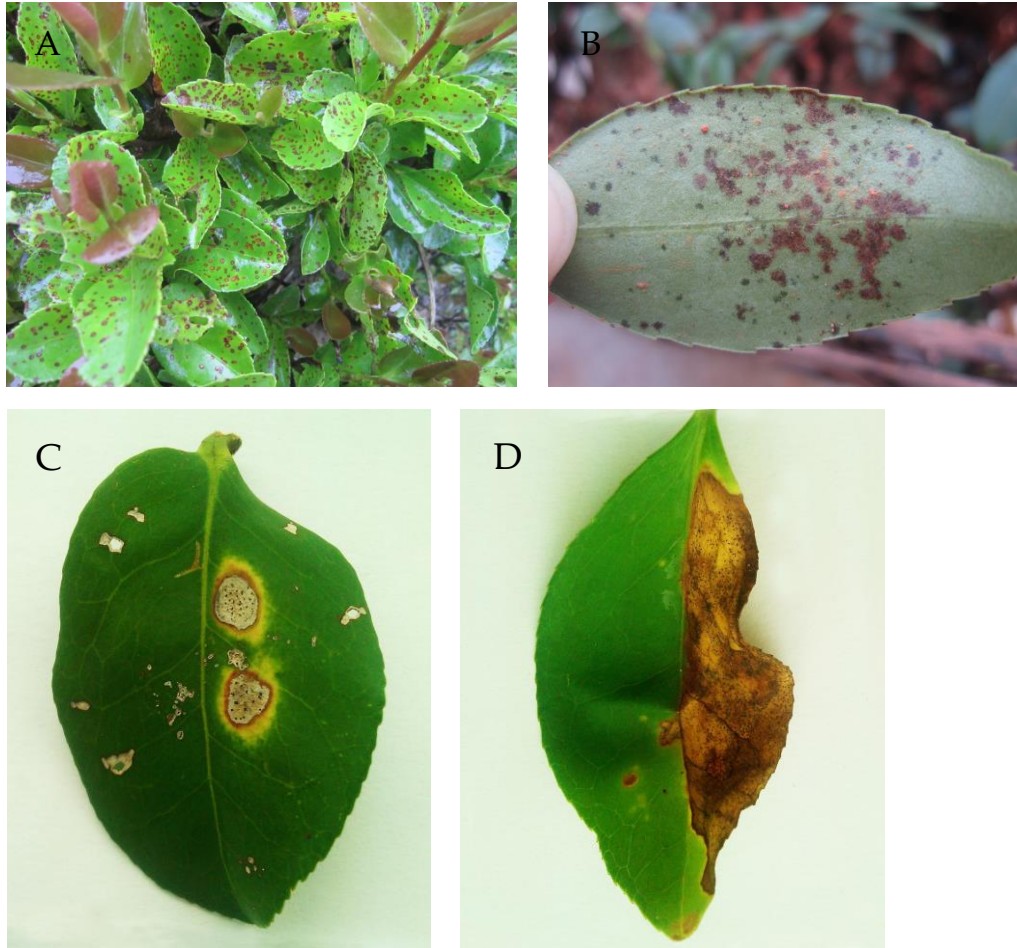

**Figure 1.** Field symptoms of tea-oil camellia anthracnose. (**A**) Initial symptoms with small brown spots. (**B**) The reverse side of initial diseased leaf. (**C**) Big lesion with black particles. (**D**) Deciduous leaf with a big lesion coalesced by small spots.

### 3.2. Fungal Isolation

Based on colony and conidia morphology, 51 *Colletotrichum* isolates were obtained from the collected leaf samples with symptoms of anthracnose. Of all the isolates, 21 isolates (41.18%) were collected from Wenchang samples, 17 isolates (33.33%) were collected from Qiongzhong samples, and 13 isolates (25.49%) were collected from Wuzhishan samples.

### 3.3. Cultural and Morphological Characteristics

All 51 fungal isolates were recognized as *Colletotrichum* species based on the colony and conidia characteristics. Distinct morphological characteristics, including the colony characteristics, conidial shape, and growth rate of the mycelium, were initially observed. Based on a comprehensive morphological characteristics analysis, all isolates characterized in this study were divided into four groups (Table 5). The isolates of Group 1 had white, mostly less fluffy, few flat colonies with less sporulation (Figure 2A,B). Conidia were cylindrical, 12.40 ± 1.29 μm × 4.51 ± 0.61 μm (Figure 2C). The appressoria in the slide culture were avoid or irregular, 9.20 ± 0.94 μm × 5.62 ± 0.45 μm, brown to dark brown (Figure 2D). The isolates of Group 2 had white to grey to dark grey colonies with medium sporulation and there were orange conidial masses near the central point (Figure 2E). The PDA was stained to light brown or dark brown on the reverse side (Figure 2F). The conidia were fusiform, 13.02 ± 0.85 μm × 4.63 ± 0.35 μm (Figure 2G). The appressoria were spherical to cylindrical, 7.51 ± 0.99 μm × 5.62 ± 0.61 μm, brown to dark brown (Figure 2H). The isolates of Group 3 had white, dense fluffy colonies with whitish-grey, sparse with floccose aerial mycelia in the center of the plate (Figure 2I,J). The conidia were fusiform, 13.65 ± 0.92 μm × 5.22 ± 0.41 μm, with only a few being one-septate (Figure 2K). The appressoria were clavate or irregular, 13.50 ± 0.75 μm × 7.15 ± 1.32 μm, brown to dark brown (Figure 2L). The colony on PDA of Group 4 was similar to that of Group 2, but the colony was grey and light brown on the reverse side with medium to high sporulation (Figure 2M,N). The conidia were fusiform, 13.22 ± 0.69 μm × 4.55 ± 0.34 μm (Figure 2O). The appressoria were similar to those of Group 2 in shape and size (Figure 2P). As to the rate of mycelial growth, the isolates of Group 2 and Group 4 were generally faster than those of the other two groups. The isolates of Group 3 had the slowest average mycelial growth rate (Table 5). Detailed cultural and morphological characteristics of the 51 isolates are described in Table S2. Combining the morphological characteristics and phylogenetic analysis, it was determined that the Group 1 and Group 3 isolates belonged to *C. fructicola* and *C. cordylinicola*, respectively, while the Group 2 and Group 4 isolates belonged to *C. siamense*.

### 3.4. Pathogenicity Test

For the leaf infection tests, all isolates were able to infect tea-oil camellia leaves (Table 6). Diseased leaves developed symptoms five days after inoculation and yellow halos occurred near the inoculated sites. Later, light brown to dark brown necrotic lesions, round or subcircular in shape, were present in all leaves three weeks after inoculation (Figure 3). Lesions produced by different isolates were comparable in size to those surveyed in fields. Isolates of different groups showed high diversity in virulence. *C. fructicola* (Group 1) showed the highest infection incidence (100%), while the infection incidence of Group 4 (75%) was lower than that of Groups 2 (95.5%) and 3 (88.3%). In the fruit infection assays, all isolates, except yc03 and yc14, showed generally weak virulence to tea-oil camellia fruits (Table 6). The isolates of Group 3 did not infect fruit at all. Diseased fruits had an irregular dark brown rot lesions on the surface. The control leaves and fruits did not show symptoms (Figure 3). Using the abovementioned method, all the isolates were consistently obtained from the infected leaves and fruits. All the isolates were confirmed by morphological examination and DNA sequencing. All the *Colletotrichum* isolates were the pathogens of tea-oil camellia anthracnose by Koch's postulates.

**Table 5.** Morphological characterization of *Colletotrichum* isolates from tea-oil camellia anthracnose.

| Groups | Isolates | Colony Characteristics | Conidia | | | Appressorium | | | Growth Rate (mm/day) |
|---|---|---|---|---|---|---|---|---|---|
| | | | Length (μm) | Width (μm) | Shape | Length (μm) | Width (μm) | Shape | |
| 1 | yc01–yc18 | White to pale, less fluffy mycelia, reverse light yellowish, less sporulation | 12.40 ± 1.29 (10.00–15.70) | 4.51 ± 0.61 (3.20–6.00) | Cylindrical | 9.20 ± 0.94 (8.00–10.70) | 5.62 ± 0.45 (4.20–6.00) | Ovoid or irregular | 9.60 ± 0.50 (8.20–10.70) |
| 2 | yc19–yc29 | White to grey to dark grey aerial mycelia, with orange visible conidial masses, reverse dark brownish, fast growing | 13.02 ± 0.85 (11.20–14.40) | 4.63 ± 0.35 (4.00–5.20) | Fusiform | 7.51 ± 0.99 (6.00–9.10) | 5.62 ± 0.61 (4.60–6.20) | Spherical to cylindrical | 11.40 ± 0.60 (10.90–13.00) |
| 3 | yc30–yc45 | White, dense fluffy mycelia with floccose aerial mycelia in center, reverse slightly greenish to brownish | 13.65 ± 0.92 (11.80–15.40) | 5.22 ± 0.41 (4.60–6.30) | Fusiform | 13.50 ± 0.75 (12.50–14.70) | 7.15 ± 1.32 (4.80–8.00) | Clavate or irregular | 8.56 ± 0.49 (8.10–9.27) |
| 4 | yc46–yc51 | Cottony, dense grey aerial mycelium, with orange visible conidial masses, reverse slightly brownish, fast growing | 13.22 ± 0.69 (12.00–14.40) | 4.55 ± 0.34 (4.00–5.10) | Fusiform | 7.83 ± 0.75 (6.20–8.50) | 6.10 ± 0.77 (4.50–6.80) | Spherical to cylindrical | 11.14 ± 0.86 (10.10–12.70) |

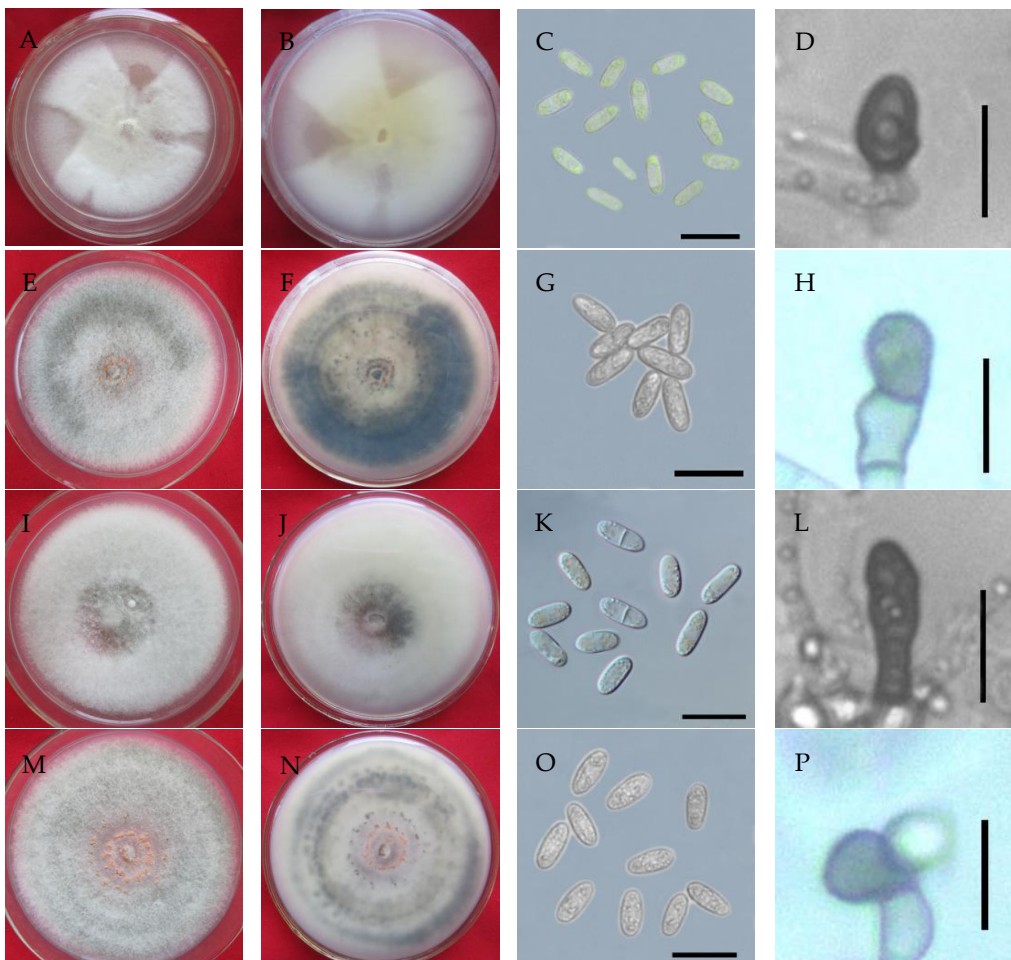

**Figure 2.** Morphological characteristics of representative isolates from different morphological groups isolated from tea-oil camellia anthracnose. (**A,E,I,M**) Morphological characteristics of colonies on the upper sides incubated on the PDA of isolates yc03, yc19, yc30, and yc51, respectively. (**B,F,J,N**) Reverse sides on PDA. (**C,G,K,O**) Morphological characteristics of the conidia. (**D,H,L,P**) Morphological characteristics of the appressoria. Scale bar (**C,G,K,O**) = 20 μm. Scale bar (**D,H,L,P**) = 10 μm.

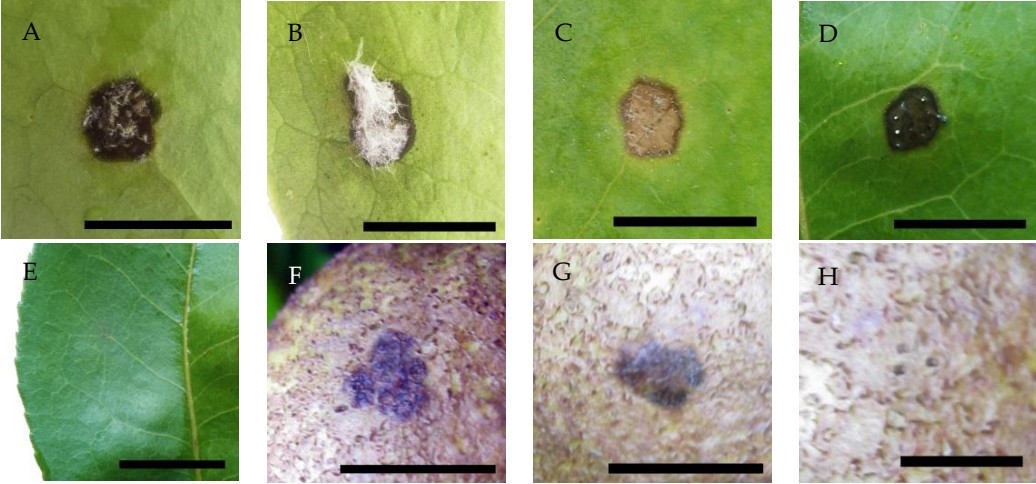

**Figure 3.** Symptoms on tea-oil camellia 15 days after inoculation. (**A**) cy03 leaf; (**B**) cy19 leaf; (**C**) cy30 leaf; (**D**) cy51 leaf; (**E**) control leaf; (**F**) yc03 fruit; (**G**) yc14 fruit; and (**H**) control fruit. Scale bar = 1 cm.

**Table 6.** Infection incidence of the *Colletotrichum* isolates on tea-oil camellia.

| Groups | Isolate No. | Leaf | Fruit | Groups | Isolate No. | Leaf | Fruit |
|---|---|---|---|---|---|---|---|
| 1 | yc01 | 100 | 20 | 2 | yc27 | 100 | 0 |
| 1 | yc02 | 100 | 6.7 | 2 | yc28 | 100 | 0 |
| 1 | yc03 | 100 | 53.3 | 2 | yc29 | 100 | 0 |
| 1 | yc04 | 100 | 0 | 3 | yc30 | 100 | 0 |
| 1 | yc05 | 100 | 13.3 | 3 | yc31 | 100 | 0 |
| 1 | yc06 | 100 | 6.7 | 3 | yc32 | 60 | 0 |
| 1 | yc07 | 100 | 0 | 3 | yc33 | 66.7 | 0 |
| 1 | yc08 | 100 | 10 | 3 | yc34 | 80 | 0 |
| 1 | yc09 | 100 | 0 | 3 | yc35 | 100 | 0 |
| 1 | yc10 | 100 | 0 | 3 | yc36 | 100 | 0 |
| 1 | yc11 | 100 | 3.3 | 3 | yc37 | 63.3 | 0 |
| 1 | yc12 | 100 | 3.3 | 3 | yc38 | 100 | 0 |
| 1 | yc13 | 100 | 0 | 3 | yc39 | 100 | 0 |
| 1 | yc14 | 100 | 50 | 3 | yc40 | 76.7 | 0 |
| 1 | yc15 | 100 | 6.7 | 3 | yc41 | 66.7 | 0 |
| 1 | yc16 | 100 | 0 | 3 | yc42 | 100 | 0 |
| 1 | yc17 | 100 | 0 | 3 | yc43 | 100 | 0 |
| 1 | yc18 | 100 | 3.3 | 3 | yc44 | 100 | 0 |
| 2 | yc19 | 100 | 6.7 | 3 | yc45 | 100 | 0 |
| 2 | yc20 | 100 | 0 | 4 | yc46 | 63.3 | 10 |
| 2 | yc21 | 100 | 0 | 4 | yc47 | 66.7 | 0 |
| 2 | yc22 | 100 | 0 | 4 | yc48 | 60 | 10 |
| 2 | yc23 | 73.3 | 10 | 4 | yc49 | 60 | 0 |
| 2 | yc24 | 76.7 | 0 | 4 | yc50 | 100 | 6.7 |
| 2 | yc25 | 100 | 0 | 4 | yc51 | 100 | 0 |
| 2 | yc26 | 100 | 6.7 | — | — | — | — |

### 3.5. Multilocus Phylogenetic Analysis

According to the comparison between obtained ITS sequences and those in GenBank using BLAST searches, all 51 isolates belong to *Colletotrichum* species. By applying the BLASTN analysis of the obtained six gene sequences in the GenBank database, all 51 *Colletotrichum* isolates were confirmed as belonging to the CGSC. BLASTn searches showed that the ITS, TUB, ACT, GAPDH, CHS-1, and SOD2 sequences of Group 1 were 97.5% to 100% similar to *C. fructicola* ICMP18613, the ITS, TUB, ACT, GAPDH, CHS-1, and SOD2 sequences of Group 2 were 98.40% to 99.83% similar to *C. siamense* ICMP18121, the ITS, TUB, ACT, GAPDH, CHS-1, and SOD2 sequences of Group 3 were 97.06% to 100% similar to *C. cordylinicola* ICMP18579, and the ITS, TUB, ACT, GAPDH, CHS-1, and SOD2 sequences of Group 4 were 98.43% to 99.83% similar to *C. siamense* ICMP18121.

The topologies of the phylogenetic trees derived from the Bayesian analysis and maximum likelihood method were similar. In this study, 51 isolates in the multilocus phylogenetic tree were grouped into three clades (Figure 4). All the Group 1 isolates (yc01–yc18) belonged to the first clade together with *C. fructicola* CBS 125,395 from *T. cacao* in Panama and ICMP18613 from *L. sinuatum* in Israel. The Group 2 (yc19–yc29) and Group 4 (yc46–yc51) isolates clustered together with *C. siamense* ICMP18121 from *Dioscor earotundata* in Nigeria and ICMP12567 from *Perseaa mericana* in Australia, forming the second clade. Finally, the Group 3 (yc30–yc45) isolates clustered together with *C. cordylinicola* ICMP18579 from *C. fruticosa* in Thailand. Combined with the analysis of morphology, pathogenicity, and DNA phylogenetic analysis, the causal pathogens of tea-oil camellia anthracnose in our study were confirmed as *C. fructicola*, *C. siamense*, and *C. cordylinicola*. The most common species in this study was *C. fructicola*, accounting for 35.29%, followed by *C. siamense* at 33.3% and *C. cordylinicola* at 31.37%.

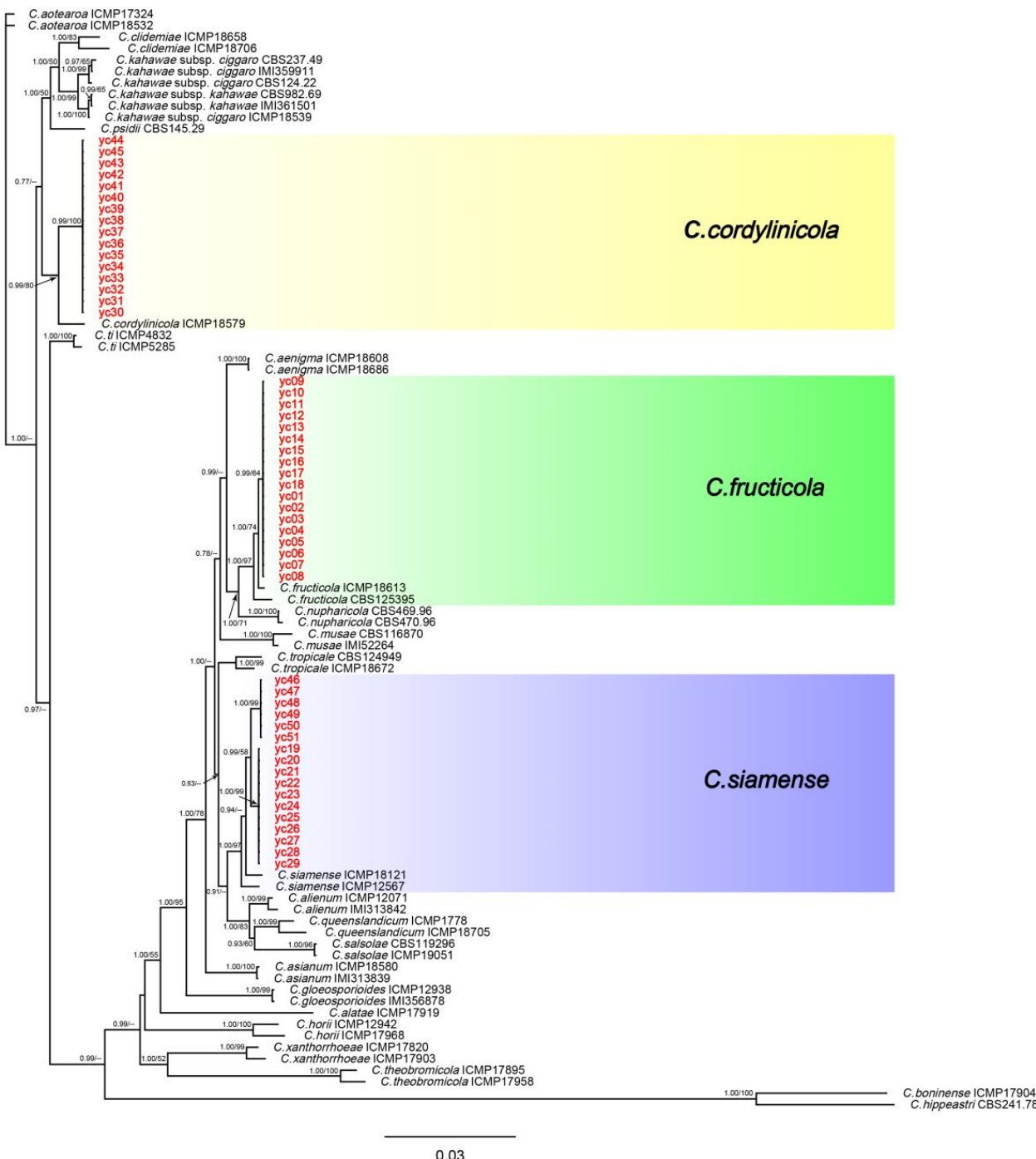

**Figure 4.** A phylogenetic tree of *Colletotrichum* species in the CGSC using Bayesian inference based on six genes (ITS, TUB, GAPDH, ACT, CHS-1, and SOD2). Bayesian posterior probability values (BI ≥ 0.70) and ML bootstrap support values (ML ≥ 50%) were indicated near the nodes (BI/ML). The isolates obtained in the present study are shown in red and bold.

## 4. Discussion

The correct identification of the causal agents of plant diseases is crucial to further clarify disease epidemiology and to establish efficient prevention and control techniques [18]. With the development of fungal species identification, more and more fungal pathogens have been identified successfully using molecular skills, including the sequencing and analysis of specific genes or loci [49–51]. Recently, many species in the CGSC with comparable morphological characteristics have been identified worldwide [52]. A variety of taxa from

this species complex have been differentiated using multilocus phylogenetic analysis based on multiple genes, including ITS, ACT, TUB2, CAL, SOD2, HIS, GS, and GAPDH [21]. Traditional fungal identification methods, including morphological characteristics and pathogenicity combined with multilocus phylogenetic analysis, proved to be an efficient way to identify *Colletotrichum* species. In this study, we obtained 51 *Colletotrichum* isolates from diseased tea-oil camellia leaves. All the isolates were characterized morphologically, such as the colonial color and texture, the hyphal growth rate, and conidial and appressorial shape and size. There were minor differences among the isolates regarding the morphology of the colonies, conidia, and appressoria, but it was difficult to distinguish these isolates to the species level via these differences. Thus, multilocus phylogenetic analyses were employed to separate these taxa. In the six-gene combined phylogeny in this study, the species relationships were well-defined and the species infecting tea-oil camellia in Hainan were recognized as *C. siamense*, *C. fructicola*, and *C. cordylinicola*. Accordingly, multiple anthracnose pathogen resistance will be an important aim of tea-oil camellia breeding programs.

A previous report showed that *C. fructicola* is a non-host-specific pathogen infecting fruit, vegetables, and economic crops, including apple (*Malus pumila*) [37], sugarcane (*Saccharum officinarum*) [38], blueberry (*Vaccinium corymbosum*) [53], and *Camellia yuhsienensis* [54]. *C. fructicola* was described as a common species existing on the leaves of tea-oil camellia [35]. The occurrence of *C. fructicola* in all the three locations in this study indicates its wide geographic spread in Hainan. *C. siamense* was obtained from Qiongzhong and Wuzhishan, while *C. cordylinicola* was isolated only from Wenchang. Interestingly, *C. siamense* showed broad variation in cultural appearance [21]. Although 11 isolates of *Colletotrichum* Group 2 and six isolates of *Colletotrichum* Group 4, isolated from different locations, were slightly different on the morphology of colony and conidia, they were all identified to be the species of *C. siamense* by phylogenetic analyses, which showed the unreliability and inconsistency of *Colletotrichum* identification methods relying only on colonial and morphological characteristics because fungal colony and morphological characteristics are affected by environmental factors [38]. In the present study, a new *Colletotrichum* species, *C. cordylinicola*, was reported causing anthracnose on tea-oil camellia in China. *C. cordylinicola* was first reported causing the disease of *C. fruticosa* in Thailand [55] and was also one of the causal pathogens of mango anthracnose in China [56]. According to our understanding, *C. cordylinicola* was reported to infect tea-oil camellia for the first time worldwide in this study. Accordingly, associated research on the population biology, host range, and fungicide sensitivity of *C. cordylinicola* in Hainan should be carried out to develop appropriate management measures.

The disease symptoms of the leaves described in the present study were similar to those in previous reports [27]. However, what merits our attention is that the disease seldom infects fruits. Ye et al. reported that tea-oil camellia fruit can be infected by anthracnose in Guangxi Province [57]. Pathogenicity tests of the 51 *Colletotrichum* isolates showed that all species were pathogenic to leaves. Previous report showed that different *Colletotrichum* species had variable virulence [58]. *Colletotrichum fructicola* showed stronger virulence on leaf than the other three Groups. *C. siamense* in Group 4 showed a medium level of aggressiveness. Most *Colletotrichum* isolates had weak aggressiveness on tea-oil camellia fruits. The different degree of fruit maturity may be associated with low infection incidence [59]. Considering the variable resistance of different tea-oil camellia cultivars to *Colletotrichum* species, future research should focus on the virulence potential of the *Colletotrichum* isolates on more cultivars of tea-oil camellia.

## 5. Conclusions

Based on the survey of three tea-oil-camellia-cultivating areas in Hainan, anthracnose was the most prominent disease, and it severely affected the growth vigor and yield of tea-oil camellia. In this study, the etiology of anthracnose on tea-oil camellia in Hainan Province was systematically studied for the first time. A total of 51 *Colletotrichum* isolates from anthracnose-infected tea-oil camellia leaves were collected and characterized. All the

isolates were characterized based on colonial characteristics, and conidial and appressorial morphology. A multilocus phylogenetic analysis based on ITS, TUB, GAPDH, ACT, CHS-1, and SOD2 was conducted using both the BI and ML methods. Pathogenicity tests on detached tea-oil camellia leaves and fruits were performed for all isolates. The causal agents of tea-oil camellia anthracnose in Hainan were identified as two reported species, *C. fructicola* and *C. siamense*, as well as one new species, *C. cordylinicola*, which demonstrates that tea-oil camellia anthracnose can be caused by multiple *Colletotrichum* species in Hainan. To our knowledge, *C. cordylinicola* is described here as a causal agent of tea-oil camellia globally for the first time. This research paper supplied a description of the morphological characteristics, pathogenicity, and molecular characteristics of the 51 *Colletotrichum* isolates obtained from tea-oil camellia in Hainan, China. Our study enriches the etiology of anthracnose on tea-oil camellia and has an important role in disease monitoring and creating management strategies for future application.

**Supplementary Materials:** The following supporting information can be downloaded at: https://www.mdpi.com/article/10.3390/f14051030/s1, Table S1: GenBank accession numbers of the *Colletotrichum* isolates from this study; Table S2: Mrophological characteristics of 51 *Colletotrichum* isolates obtained in this study.

**Author Contributions:** Conceptualization, H.Z. and C.H.; methodology, H.Z.; software, H.Z.; validation, H.Z. and C.H.; formal analysis, H.Z.; investigation, H.Z.; resources, H.Z.; data curation, H.Z.; writing—original draft preparation, H.Z.; writing—review and editing, H.Z.; visualization, C.H.; supervision, C.H.; project administration, H.Z.; funding acquisition, C.H. All authors have read and agreed to the published version of the manuscript.

**Funding:** This research was funded by the Major Research Development Program of Hainan Province, grant number ZDYF2019072.

**Institutional Review Board Statement:** Not applicable.

**Informed Consent Statement:** Not applicable.

**Data Availability Statement:** All sequence data are available in the NCBI GenBank following the accession numbers in the manuscript.

**Acknowledgments:** The authors are grateful to Fangluan Gao, a researcher of Fujian Agriculture and Forestry University, for his great help in constructing the phylogenetic tree.

**Conflicts of Interest:** The authors declare no conflict of interest.

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
