# Peer review of "Identification and Characterization of Colletotrichum Species Causing Tea-Oil Camellia (Camellia oleifera C.Abel) Anthracnose in Hainan, China"

_forests, doi:10.3390/f14051030_

Round 1

Reviewer 1 Report

Paper Titled “Identification and Characterization of Colletotrichum SpeciesCausing Tea-Oil Camellia (Camellia oleifera) Anthracnose in Hainan, China” dealing with anthracnose disease of tea-oil camellia (Camellia oleifera). The paper is a straightforward, well written, nicely structured with good presentation, I recommend for publication with minor correction.

There are some changes or corrections, which are:

Page1, L15: change the word ‘locations’ to ‘sites’

Page1, L16: Change the sentence as “Initially, all the 51 isolates of Colletotrichum were characterized morphologically, followed by DNA sequencing of 6 gene regions, including internal transcribed spacer region (ITS1-5.8S-ITS2 = ITS) of nuc ribosomal DNA, chitinsynthase (CHS-1), β-tubulin (TUB), actin (ACT), glyceraldehyde-3-phosphate dehydrogenase (GAPDH), and manganese-superoxide (SOD2)”. Please check the abbreviations for these genes.

Page1, L20: change the word ‘morphology analyses’ with morphological characterization’

Page1, L20-21: replace the sentence ‘all the isolates were identified as Colletotrichum fructicola, C. siamense, and C. cordylinicola’ with ‘the fungal isolates were identified, representing three Colletrotrichum species: C. fructicola, C. siamense, and C. cordylinicola, respectively’.

Page1, L22-23: (Hainan Local) this is the name of the cultivar? If not, please provide the variety name.

Page1, L24: Write the complete generic name at the start of a sentence “Colletotrichum”.

Page2, L45: add authority when taxonomic rank (Family, genus, species) for the first time in the text.

Page2, L70: Please explain these abbreviation GAPDH, ACT, CHS, TUB, etc.

Page2, L73: Replace the sentence ‘by multiple gene phylogenetic analyses based on eight genes’ with ‘using multi-loci (eight gene) phylogeny [21]’.

Page2, L74-75. After reference [21] there is directly reference [26], no reference between 22-25. ??.

Page2, L87: “caused by” ??. I think it would be ‘symptoms of anthracnose on tea oil camelia…….”.

Page7, L187: How you make the concatenated dataset? What was the criteria of arranging the gene se  quence?

Page19, L292: Figure 3, please add the scale bar.

Page21, L335: replace ‘spp. to the species level’ with ‘Colletotrichum species’.

Page21, L341: change the word ‘required’ with ‘employed’.

Overall, the quality of language is good.

Reviewer 2 Report

The manuscript titled “Identification and Characterization of Colletotrichum Species Causing Tea-Oil Camellia (Camellia oleifera) Anthracnose in Hainan, China” is devoted to the identification and characterization of three Colletotrichum species (C. fructicola, C. siamense, and C. cordylinicola) causing anthracnose on tea-oil camellia in Hainan based on their morphology, virulence, and molecular features. It represents and valuable study, where C. cordylinicola is generally identified for the first time on tea-oil camellia.

The paper is well-written and the results obtained support the conclusions raised by the Authors. However, there are some minor changes to be addressed, especially in Results, where authors have to filter text and delete redundant sentences that should be written or are already written in the Materials and Methods section:

Comments to authors

Abstract

Lines 13-14: Rephrase the sentence as follows: “In this study, fifty-one Colletotrichum spp. isolate was obtained from the symptomatic samples of tea-oil camellia, collected from three production sites located in Hainan.”

Line 16: Replace ‘firstly’ with ‘primarily’

Lines 19-20: Rephrase the sentence as follows: “By combining morphological analyses with multilocus sequence analysis (MLSA) based on the six genes, tested isolates were identified as Colletotrichum fructicola, C. siamense, and C. cordylinicola.

Line 26: Replace ‘comprehension’ with ‘understanding’

Line 28: I suggest replacing ‘multi-locus phylogeny’ with ‘multilocus sequence analysis’

Introduction

Line 32: Rephrase the sentence as follows: “Tea-oil camellia (Camellia oleifera Abel.), originating from China, is the main crop used for the production of edible oil in this country.”

Line 38: diseases

Line 40: Correct the sentence as follows: “Therefore, tea-oil camellia is not only capable of improving human health but also is...”

Line 52: of strawberry fruit

Line 64-66: “It was difficult to identify these Colletotrichum spp. to the species level only using morphological and biological characteristics.”

The sentence can be deleted.

Line 66-67: Rephrase the sentence as follows: “Therefore, molecular approaches are being increasingly applied for their identification [14,18].”

Line 70: Write a full explanation for GAPDH, ACT, CHS, and TUB, like in the abstract section

Lines 87-88: symptoms of anthracnose

Materials and Methods

Line 103: The three plantations are located in...

Line 118: Please write the number of the obtained Colletotrichum isolates and move Table 4 with provided isolates codes, localities and years of isolation from Results to Materials and Methods.

Line 122: ...color and zonation of each isolate...

Line 132: ....for each of the (write number of isolates) isolates

Line 147-148: Please explain how was the infection incidence calculated

Line 158: Seems like a total reaction volume was 26.5 μL. Please check

Line 171: were sent to Shanghai Sangon Biotech Co. Ltd.

Line 172: Replace ‘senses’ with ‘directions’

Line 176: GenBank accession numbers of the Colletotrichum isolates from this study

Table 2 can be deleted and the accession numbers can be written in the form of text or the table can be moved to the supplementary material

Lines 178-181: “According to the comparison between obtained rDNA-ITS sequences and those in GenBank using BLAST searches, all the 51 isolates belong to Colletotrichum species. By applying the BLASTN analysis of the obtained six gene sequences in the GenBank database, all the 51 Colletotrichum isolates were confirmed as belonging to the CGSC.”

The sentences should be moved to the Results section.

Line 182: Write a number of different reference species

Line 187: Please write the final length of the concatenated sequences

Line 200: Write strain codes for outgroup strains

Results

Line 208: delete ‘in a year’

Lines 205-206: “During 2017–2021, the anthracnose of tea-oil camellia in three major districts of Hainan Province was surveyed.”

This is Materials and Methods. The sentence can be deleted.

Lines 217-220: “The disease symptoms of the leaves described in the present study were similar to previous reports [27]. However, what merits our attention is that the disease seldom infects fruits. Ye et al. reported that tea-oil camellia fruit can be infected by anthracnose in Guangxi Province [49].”

This is not results, but discussion. The sentences can be moved to the Discussion section.

Line 225-226: Rephrase the sentence as follows: “Based on colony and conidia morphology, 51 Colletotrichum isolate was obtained from the collected leaf samples with symptoms of anthracnose (Table 4 – add new Table no.).”

Line 226: “from 185 leaf samples with symptoms of anthracnose”

Write this information in Materials and Methods section.

Lines 226-227: “All the isolates were purified using single-spore cultures. Detailed information of the isolates is shown in Table 4.”

The first sentence is Materials and Methods. The sentences can be deleted.

Lines 230-232: “Some endophytic or minor microbes, including Phomopsis, Alternaria, and Penicillium, were also obtained, but they were not further investigated. All isolates were selected for further research on morphological characteristics, pathogenicity test, and molecular identification.”

The sentences can be deleted.

Line 272: “All 51 Colletotrichum isolates were used for the pathogenicity tests.”

This is Materials and Methods. The sentence can be deleted.

Line 282: “which explained the low infection incidence of fruit in the field surveys.”

This part of the sentence can be deleted.

Lines 282-283: “The isolates of Group 3 did not infect fruit at all.”

In Table 6 their pathogenicity was scored as ‘+’ which represents infection incidence below 50%. You should explain the used scale for scoring disease intensity in Materials and Methods. Perhaps you should consider modifying scores in a way that you have more levels including a ‘zero infection level’ or ‘healthy’ for plants without developed symptoms. From the results presented in this manner, one can’t resolve that some isolates weren’t pathogenic on fruits at all. I suggest using some scale where scores are not defined with ‘+’ or  ‘-‘, but with scores that can be transferred to numbers representing the percent of disease intensity.

Line 295: Perhaps you can also add the results of NCBI BLASTn analysis, showing the percent identity of tested isolates with sequences of strains deposited into the GenBank for each of the four detected groups of isolates based on all six sequenced genes.

Lines 295-304: The whole paragraph should be moved to the Material and Methods section.

Table S1: Morphological characteristics

Please refer to Table S1 at the appropriate place in the text

Discussion

The authors should discuss results related to the virulence potential of the three detected Colletotrichum species on leaves and fruits.

Line 367-372: The whole paragraph can be moved to Conclusions and replace the text written in lines 374-389

Reviewer 3 Report

The paper is interesting, rather well written (yet, English should be improved!) and the results are based on carefully performed research.

English should be improved, but see my comments/corrections in the reviewed manuscript.
